# Risk Factors for Bacteremia and Its Clinical Impact on Complicated Community-Acquired Urinary Tract Infection

**DOI:** 10.3390/microorganisms11081995

**Published:** 2023-08-02

**Authors:** Manuel Madrazo, Ian López-Cruz, Laura Piles, Silvia Artero, Juan Alberola, Juan Alberto Aguilera, José María Eiros, Arturo Artero

**Affiliations:** 1Doctor Peset University Hospital, University of Valencia, 46017 Valencia, Spain; manel.madrazo@gmail.com (M.M.); ilopezcruz5@gmail.com (I.L.-C.); laurapilesroger@gmail.com (L.P.); juan.alberto.aguilera1994@gmail.com (J.A.A.); arturo.artero@uv.es (A.A.); 2Gregorio Marañón University Hospital, 28007 Madrid, Spain; silvia.artefull@gmail.com; 3Rio Hortega University Hospital, Universidad de Valladolid, 47012 Valladolid, Spain; eiros@med.uva.es

**Keywords:** risk factors, bacteremia, urinary tract infections

## Abstract

Bacteremia has been associated with severity in some infections; however, its impact on the prognosis of urinary tract infections (UTIs) is still disputed. Our goal is to determine the risk factors for bacteremia and its clinical impact on hospitalized patients with complicated community-acquired urinary tract infections. We conducted a prospective observational study of patients admitted to the hospital with complicated community-acquired UTIs. Clinical variables and outcomes of patients with and without bacteremia were compared, and multivariate analysis was performed to identify risk factors for bacteremia and mortality. Of 279 patients with complicated community-acquired UTIs, 37.6% had positive blood cultures. Risk factors for bacteremia by multivariate analysis were temperature ≥ 38 °C (*p* = 0.006, OR 1.3 (95% CI 1.1–1.7)) and procalcitonin ≥ 0.5 ng/mL (*p* = 0.005, OR 8.5 (95% CI 2.2–39.4)). In-hospital and 30-day mortality were 9% and 13.6%, respectively. Quick SOFA (*p* = 0.030, OR 5.4 (95% CI 1.2–24.9)) and Barthel Index <40% (*p* = 0.020, OR 4.8 (95% CI 1.3–18.2)) were associated with 30-day mortality by multivariate analysis. However, bacteremia was not associated with 30-day mortality (*p* = 0.154, OR 2.7 (95% CI 0.7–10.3)). Our study found that febrile community-acquired UTIs and elevated procalcitonin were risk factors for bacteremia. The outcomes in patients with bacteremia were slightly worse, but without significant differences in mortality.

## 1. Introduction

Community-acquired bacteremia is a major cause of morbidity and mortality worldwide [1,2,3]. Urinary tract infection (UTI) is among the most common bacterial infections in the community [4] and it is the most common source of bacteremia. UTI is a widespread condition that can affect individuals of all ages, but it has a particularly significant impact on elderly patients [5]. UTI is more commonly found in women, but as men age, their likelihood of experiencing UTI increases. Aging is associated with increased risk factors for UTI, including factors such as anatomical abnormalities and weakened immune responses. Indeed, men over the age of 80 have a comparable infection rate to women [6]. Complicated UTIs (cUTIs) are a subset of UTIs that occur in the setting of pre-existing structural or functional abnormalities of the urinary tract, resulting from a broad range of clinical syndromes [7]. Bacteremia from urinary source has a lower mortality than bacteremia from other sources of infection [8,9] and its influence as an independent risk factor on the prognosis is still under debate. Among young women with uncomplicated UTIs, the presence of bacteremia did not show any association with the severity of the infection or its outcomes [10], while other studies conducted on the clinical impact of bacteremia on the prognosis of complicated UTIs in adults yielded conflicting results [11,12].

Bacteremia has been reported in 20% to 31% of cases of acute pyelonephritis [10,13,14] and in 38% to 69% severe sepsis or septic shock [15]. In addition, bacteremia was the only way to establish etiology in 7% to 11.6% of complicated UTI when urine cultures were negative or contaminated [16]. This was especially useful for those cases receiving antibiotics at the moment of hospitalization, in which blood cultures have additional diagnostic value over urine cultures. Accordingly with the microbiology of UTIs, Gram-negative organisms predominate in bacteremia from urinary source [17], mainly Enterobacterales, which present a growing problem of antibiotic resistance, not only in nosocomial infections but also in community-acquired urinary tract infections [18].

Predictive factors for bacteremia may be relevant for clinical management. Although several studies have studied risk factors for urinary bacteremia, these were retrospective [19,20,21,22] or analyzed only certain types of UTI, such as those on elderly patients [23,24], catheter-associated UTI [21], or febrile UTI in the emergency department [22]. Therefore, there is a need to know the risk factors and clinical significance of bacteremia in patients with complicated community-acquired UTI who require hospital admission.

Our objective for this study was to determine the risk factors for bacteremia and its clinical impact on hospitalized patients with complicated community-acquired urinary tract infection.

## 2. Materials and Methods

Prospective observational study of patients admitted to a university hospital diagnosed with community-acquired UTI, from January 2017 to December 2021. The diagnosis of UTI was first made by the attending physician in the hospital emergency department. Subsequently, it was further confirmed by the physician in charge following the patient’s admission. This confirmation was based on a meticulous assessment of the patient’s medical history, thorough physical examination, and comprehensive utilization of laboratory and microbiological diagnostic procedures. Cases with a clinical syndrome compatible with any other condition after reviewing the case were excluded. Patients with asymptomatic bacteriuria, which is defined as isolation of a specified quantitative count of bacteria in an appropriately collected urine specimen from an individual without symptoms of UTI [25], were also excluded. Other exclusion criteria were: nosocomial infections (considered as UTI starting after hospital admission [26]), UTI cases transferred from the Intensive Care Unit, patients with no blood culture and those who refused to participate (see Figure 1).

All patients signed an informed consent before being included in the study. Epidemiological and clinical variables were collected by the authors following a protocol [27]. All the cases were reviewed by two independent researchers before being included in the study.

The term bacteremic UTI referred to cases where a compatible microorganism was detected in the blood culture during the initial 48 h of hospitalization [22]. Coagulase-negative Staphylococcus found in blood cultures was regarded as a contaminant, and such cases were categorized within the negative blood culture group [20]. The SOFA and qSOFA scales were employed in accordance with their original definitions [28], and their measurements were taken within 24 h of admission at the Emergency Department (ED). In this study, Acute UTI was classified as complicated if the patient exhibited structural or functional urinary tract abnormalities, indicating a high probability of treatment failure and potential serious complications [29]. The study defined Community Onset Healthcare-Associated UTI (HCA-UTI) as a community-onset UTI that met any of the following criteria: (i) having been admitted to an acute care hospital for at least 48 h within the 90 days prior to the current hospital admission; (ii) receiving antimicrobial therapy within 90 days before admission; or (iii) residing in a nursing home [30]. Community-acquired infection was defined as a urinary infection in which the symptoms originated in the community, and none of the previously mentioned criteria for HCA-UTI were present [30].

The study included the following comorbidity: diabetes mellitus, which was determined by fasting plasma glucose values ≥ 126 mg/dL or glycated hemoglobin values ≥ 6.5 percent; cognitive impairment was identified according to clinical criteria for dementia or mild neurocognitive disorder as specified in the Diagnostic and Statistical Manual for Mental Disorders, Fifth Edition (DSM-5); chronic kidney disease was characterized by a reduced glomerular filtration rate of less than 60 mL/min, as determined using the CKD-EPI equation; chronic obstructive pulmonary disease (COPD) was diagnosed through spirometry, which revealed airflow limitation with a forced expiratory volume in one second/forced vital capacity (FEV1/FVC) ratio less than 0.7 or lower than the lower limit of normal. Additionally, the airflow limitation was not fully reversible after the administration of an inhaled bronchodilator.

Data on clinical manifestations were directly acquired by conducting interviews with patients and performing physical examinations. Fever was considered to be present if the patient reported a temperature of ≥38 °C at home or if it was measured in the emergency department. The laboratory evaluations encompassed coagulation testing, a complete blood count, and a comprehensive analysis of blood chemistry, which included assessments of liver and renal function, electrolyte levels, procalcitonin, and C-reactive protein levels.

The study utilized the Charlson Comorbidity Index to assess the overall burden of comorbidities. A Charlson Comorbidity Index equal to or greater than 5 indicated significant comorbidity [31].

Inadequate empirical antimicrobial therapy (IEAT) was defined as instances where an infection was not effectively treated when the causative microorganism and its antimicrobial susceptibility were already identified. This encompassed two scenarios: the absence of antimicrobial agents directed at a specific class of microorganisms and the administration of an antimicrobial agent to which the microorganism responsible for the infection was resistant [20]. Multidrug-resistant bacteria (MDRB) were defined following an international expert proposal by Magiorakos et al. [32]. MDRB was characterized by non-susceptibility to at least one agent in three or more antimicrobial categories. For Gram-negative bacteria, these categories included extended-spectrum penicillins, carbapenems, cephalosporins, aminoglycosides, and fluoroquinolones. For Gram-positive bacteria, the categories consisted of ampicillin, vancomycin, fluoroquinolones, fosfomycin, and linezolid. To identify Extended-spectrum beta-lactamase (ESBL)-producing Enterobacterales, the isolates with positive results for ESBL (typically showing reduced sensitivity to one or more of the following antibiotics: cefpodoxime, ceftazidime, cefotaxime, ceftriaxone, and aztreonam) were confirmed using the double disc synergy test (DDST). This test involved comparing the inhibition halos of a third-generation cephalosporin alone and with clavulanic acid incorporated into the discs. The increased activity of the cephalosporin in the presence of clavulanic acid indicated the production of ESBL in Gram-negative bacilli, following the guidelines of the CLSI (Clinical and Laboratory Standards Institute). Additionally, a phenotypic confirmation test was conducted on isolates suspected of having AmpC beta-lactamases. This was mainly observed in cases of resistance to third-generation cephalosporins, where a confirmatory ESBL negative test was present or intermediate sensitivity or resistance to amoxicillin with clavulanic acid and third-generation cephalosporins [33].

Sepsis was classified following the sepsis-3 criteria and identified as an acute elevation in the total SOFA score by ≥ 2 points consequent to the infection [34]. Both SOFA and quick SOFA (qSOFA) scales were employed according to their original definitions [35]. The acute physiology and chronic health evaluation classification system (APACHE II) score was used to identify illness severity at admission. The Barthel Index was employed to assess the functional status of the patients by evaluating their level of independence in activities of daily living. Quick Pitt (qPitt), a simplified quick form of the Pitt bacteremia score (PBS) using binary variables for acute severity of illness, was used according to their original definition.

Microbiological data were collected through urine and blood culture, as well as susceptibility testing. This encompassed the identification of bacteremia, determination of the causative agents of UTI through culture isolation, assessment of resistance patterns exhibited by the isolated microorganisms, and identification of cases involving polymicrobial infections. The identification of microorganisms from positive blood cultures was conducted using the Bruker MALDI Biotyper system (Beckman Coulter, Brea, CA, USA). For drug sensitivity and resistance testing, two systems were employed: the DxM MicroScan WalkAway microbiology system (Beckman Coulter, Brea, CA, USA) and The VITEK 2 system (bioMérieux, Inc., Hazelwood, MO, USA). These systems utilized microbroth dilution methods, following a combination of CLSI and EUCAST rules. Blood samples were collected in the emergency department and processed utilizing the BacT/ALERT^®^ VIRTUO™, an automated system designed for culturing and detecting microorganisms in blood. Once positive blood cultures were obtained, microorganisms were identified using the Bruker MALDI Biotyper system (Beckman Coulter, Brea, CA). For drug sensitivity and resistance tests, the DxM MicroScan WalkAway microbiology system (Beckman Coulter, Brea, CA), and The VITEK 2 system (bioMérieux) were employed, utilizing microbroth dilutions methods in accordance with a combination of CLSi and EUCAST rules.

All data for continuous variables were expressed as median and interquartile ranges. Categorical variables were reported as frequencies and percentages. Normal distribution was verified using Kolmogorov–Smirnov’s one-sample test. Quantitative variables were compared using either Student’s *t*-test or analysis of variance (ANOVA) in cases where the distribution was normal. Nonparametric data underwent analysis through a two-tailed Mann–Whitney U-test. For qualitative variables, comparisons were made using the chi-square test and Fisher’s exact test. Multivariate analysis was conducted using logistic regression, with a significance level (α) set at 0.05 for all tests. All statistical tests were two-sided. The statistical analysis was performed using the IBM SPSS version 22 for Windows.

This study received approval from the Clinical Research Ethics Committee of the Doctor Peset University Hospital and adhered to the STROBE statement.

## 3. Results

Of a total of 931 patients who were admitted to the hospital diagnosed with community-acquired UTI with a positive urine culture during the period of study, 279 had blood cultures without other exclusion criteria and were included in the study. Blood cultures were positive in 105 (37.6%) cases. The median age was 78 years, and 44.1% of patients were female. Diabetes mellitus (31.9%), dementia (22.9%), cancer (22.5%), and moderate-severe chronic kidney disease (22.2%) were the most frequent comorbidities, without significant differences between bacteremic and non-bacteremic patients. A 47.7% portion of the patients were septic on admission with SOFA ≥ 2, and 14.2% had septic shock. Out of the 61 patients with a urinary catheter, the majority had a Foley catheter. However, there were three nephrostomies in the bacteremic group and nine nephrostomies, one suprapubic catheter, and one JJ catheter in the non-bacteremic group, without statistically significant differences by ANOVA (*p* = 0.671). Other epidemiological and clinical characteristics and outcomes are given in Table 1.

Bacteremia was associated with a Charlson Index ≥ 5, acute pyelonephritis, respiratory rate ≥ 22 bpm, temperature ≥ 38 °C, qSOFA ≥ 2, SOFA ≥ 2, septic shock ≥ 2, lactate ≥ 2 mg/dl, and procalcitonin (PCT) ≥ 0.5 ng/mL by univariate analysis (see Table 1). Temperature ≥ 38 °C (OR 1.3, OR 95%, CI 1.1–1.7, *p* = 0.006) and PCT ≥ 0.5 ng/mL (OR 8.5, OR 95% CI 2.8–39.4, *p* = 0.005) were independent risk factors for bacteremia by multivariate analysis (see Table 2).

*Escherichia coli* was the most common microorganism (56.5%), followed *by Klebsiella pneumoniae* (10.1%), *Pseudomonas aeruginosa* (7.8%), *Enterococcus faecalis* (7.8%), and *Proteus mirabilis* (4.2%). *E. coli* (57%), *K. pneumoniae* (12.4%), *P. aeruginosa* (7.4%), and *E. faecalis* (6.6%) were the most common microorganisms in bacteremic patients, without statistically significant differences between bacteremic and non-bacteremic patients. The microorganisms isolated in the urine and blood were concordant in 95.7% of cases. In the 12 discordant cases, the isolated microorganisms were as follows: *K. pneumoniae* in urine and *E. coli* in blood (*n* = 3), *P. mirabilis* in urine and *K. pneumoniae* in blood *(n* = 2), *P. aeruginosa* in urine and *Klebsiella oxytoca* in blood (*n* = 1), *E. faecalis* in urine and *E. coli* in blood (*n* = 3), *E. coli* in urine and *E. faecalis* in blood (*n* = 1), *E. faecalis* in urine and *K. oxytoca* in blood (*n* = 1), and *E. faecalis* in urine and *Serratia marcescens* in blood (*n* = 1). In the bacteremic group, there was a polymicrobial infection with *E. faecalis* and *C. glabrata*, both in blood and urine cultures, and two polymicrobial infections with *P. aeruginosa* and *C. albicans*. In the non-bacteremic group, there was one case of candiduria due to *C. tropicalis* and two cases of polymicrobial infection with *C. parapsilosis* and *P. aeruginosa*. There were three UTIs due to *S. agalactiae*, one due to *S. aureus*, and one due to *S. saprophyticus*, all of which had negative blood cultures. Polymicrobial infections were more prevalent in bacteremic UTIs compared to non-bacteremic UTIs, although statistical significance was not achieved (15.2% vs. 8%, *p* = 0.060).

In-hospital mortality was 9%, with no statistically significant differences between bacteremic and non-bacteremic patients (12.4% vs. 6.9%, *p* = 0.120). In-hospital mortality was associated with Charlson Index ≥ 5 (*p* = 0.001), Barthel Index ≤40 (*p* < 0.001), HCA-UTI (*p* = 0.028), procalcitonin ≥ 0.5 ng/mL (*p* = 0.047), qSOFA ≥ 2 (*p* < 0.001), and qPitt ≥ 2 (*p* = 004). The Charlson Index ≥ 5 (OR 5.5, OR 95%, CI 1.1–30.6, *p* = 0.049) and qSOFA ≥ 2 (OR 22.6, OR 95% CI 2.2–233.2, *p* = 0.009) were independently associated with in-hospital mortality by multivariate analysis. The length of hospital stay was higher for bacteremic patients (6 days vs. 5 days, *p* = 0.002). Risk factors for 30-day mortality are shown in Table 3. Barthel Index ≤40 (OR 4.8, OR 95% CI 1.3–18.3, *p* = 0.020) and qSOFA ≥ 2 (OR 5.4, OR 95% CI 1.2–24.9, *p* = 0.030) were associated with 30-day mortality by multivariate analysis. Although the 30-day mortality was 19% in patients with bacteremia and 10.3% in those without bacteremia, no statistically significant differences were observed between both groups (OR 2.7, OR 95% CI 0.7–10.3, *p* = 0.030), see Table 3.

## 4. Discussion

In this observational study, a temperature ≥ 38 °C and a PCT ≥ 0.5 ng/mL were identified as independent risk factors for bacteremia. The outcomes in patients with bacteremia were slightly worse, but without significant differences in mortality.

In order to compare patients with UTIs with and without bacteremia, a rigorous evaluation of the medical records was conducted. Out of a total of 931 cases initially diagnosed with urinary tract infections and positive urine cultures (see Figure 1), 45.6% of them were excluded upon reviewing the medical records due to being considered asymptomatic bacteriuria associated with other pathologies causing the patients’ symptoms. Therefore, the cases included in this study represent a broad spectrum of complicated community-acquired UTIs without reducing them to specific subgroups, such as febrile UTIs.

In the current study, a higher proportion of male participants compared to females was observed, which can be attributed to the advanced age of hospitalized patients with complicated UTIs. Although the incidence of bacteremia was slightly higher in women, it did not reach statistical significance. The advanced age of the study population contributed to a substantial burden of comorbidity, with approximately one-third of cases being diabetic patients. Notably, the Charlson Comorbidity Index exhibited an association with in-hospital mortality, while the Barthel Index demonstrated an association with 30-day mortality. The study population stood out for a notable percentage of individuals with healthcare-associated UTIs (HCA-UTIs), leading to a high prevalence of prior hospitalizations, previous antimicrobial therapy, and patients originating from residential care facilities.

Fever is a clinical parameter associated with positive blood cultures that has been known for some decades [36]. In a study at the emergency department, the number of patients with febrile bacteremia was much higher than those with afebrile bacteremia from any source [37]. Temperature has been described as a risk factor for bacteremia in UTIs in previous studies [20,22]. This supports the use of simple clinical data in the study of infections. However, no other vital signs that were related to bacteremia in other studies, such as heart rate >90 rpm [20] or systolic blood pressure <100 mmHg [22,38], were found in this study.

PCT is a calcitonine precursor polypeptide whose normal synthesis is restricted to thyroid neuroendocrine cells, but it is induced in nearly all tissues by systemic inflammation, especially in those cases resulting from bacterial infections [39]. PCT is a long-known serum biomarker of bacteremia. In a meta-analysis with 58 studies and 16.514 patients with bacteremia from different sources [40], PCT had an area under the summary receiver-operating characteristic curves of 0.79, with a negative predictive value of 95% for bacteremia in the medical ward. In another study, Álvarez et al. [23] described an AUROC of 0.79 for bacteremia in a population of elderly patients with febrile UTI. According to these studies, our data supports the use of PCT as a biomarker of bacteremia in patients with UTIs. Based on our findings, we can address the study question, “Who is at risk of bacteremia in UTI?” and apply them to clinical practice by recommending that patients presenting with fever and elevated PCT levels should have blood cultures performed if they are diagnosed with a UTI.

In this study, UTIs were acquired in the community in a setting with a high rate of MDRB (31.9%) and ESBL-producing bacteria (11.8%). These findings might be attributed to the elevated proportion of HCA-UTIs (55.2%), in which the etiology of the infection resembles nosocomial infections more closely. No relationship has been found between the etiology of UTI and bacteremia, neither regarding specific microorganisms nor their characteristics such as antibiotic multi-resistance or ESBL-producing strains. It is also remarkable that a high proportion of polymicrobial infections was observed in the group of patients with bacteremic urinary tract infections (15.2%). Although this percentage was nearly double compared to non-bacteremic UTIs (8%), these differences did not reach statistical significance. Polymicrobial infections have been strongly associated with catheter-associated UTIs, in which the incidence rate can exceed 20% [41]. In our study, there were a significant number of patients with urinary catheterization, although the distribution between both groups was similar.

Comorbidities and THE functional status of patients are important determinants of prognosis in severe infections. Among the comorbidities with greater influence, some stand out, such as AIDS, liver disease, cancer, alcohol dependence, and pre-sepsis immunosuppression [42,43]. Other conditions that have been associated with a risk of severe infections include cognitive impairment, cancer, obesity, immunodeficiencies, advanced chronic kidney disease requiring hemodialysis, and malnutrition [42,43,44,45]. In our study, pre-existing diseases, individually, did not show differences in prognosis. However, a validated index such as the Charlson Comorbidity Index was associated with higher in-hospital mortality when the score was ≥5 (OR 5.5, OR 95% CI 1.1–30.6, *p* = 0.049). However, this association was not observed in 30-day mortality. This could indicate that severe comorbidity is directly related to severe infection in its hyperacute phase but plays a lesser role in subsequent recovery or medium-term mortality.

Functional status can be measured using the Barthel Index, reflecting the level of dependence on basic activities of daily living, which has been described as an independent risk factor for mortality [46,47]. Contrary to the results regarding comorbidity, severe dependence on basic activities of daily living, determined by a Barthel Index score ≤40, was associated with higher 30-day mortality (OR 4.8, OR 95% CI 1.3–18.2, *p* = 0.020). This suggests the influence of the functional state at the beginning of the infectious process on the prognosis once hospitalization is overcome.

In this study, qSOFA demonstrated a significant association with both in-hospital and 30-day mortality, surpassing bacteremia as a prognostic factor in the context of community-acquired UTIs. These findings underscore the notable impact of sepsis on short-term prognosis within the UTI population. Furthermore, since qSOFA only requires three clinical parameters for its calculation (altered mental status, low systolic blood pressure, and high respiratory rate), it is a tool that is easy to apply even in areas with limited resources. Its ease of use facilitates the swift identification of high-risk patients, enabling timely interventions and optimal resource allocation, making it a practical tool for healthcare providers in the setting of community-acquired UTIs.

Bacteremia from different sources has been related to severity and mortality in some studies [38,48], but its role in UTIs remains controversial. Some studies show an association between bacteremia from community-acquired UTIs and septic shock [12,49,50] or mortality [51], while others show no influence on outcomes [10,24,52]. A cluster analysis of different clinical phenotypes in bacteremia showed that women with UTIs had lower mortality than patients with bacteremia of other sources (19.2% vs. 27.4–44.6%, *p* < 0.001) [53]. In this study, in-hospital mortality and 30-day mortality were both higher in bacteremic groups (13.3% vs. 7.3% and 19% vs. 10.3%, respectively), but with no statistically significant differences. Also, patients with bacteremia had a longer hospital stay, according to another study [54], but not with our previous findings [24].

Another practical aspect of bacteremia is having a definitive etiological diagnosis of urinary tract infection (UTI). In the design of our study, all cases included had a positive urine culture. However, in clinical practice, it is well known that in a percentage of cases, urine cultures either cannot be obtained before the use of antibiotics or yield negative or contaminated results. In these cases, a positive blood culture allows for targeted treatment, which, in the case of a patient receiving IEAT, could improve the clinical prognosis. This could be applicable in a significant percentage of cases, given that in our series, blood cultures were positive in 37.6% of cases and IEAT was used in 24% of cases. Another aspect where a positive blood culture may have clinical implications is in cases of discordant isolates between urine and blood cultures, likely due to polymicrobial infections. In this study, 12 cases were identified where microorganisms in blood cultures differed from those isolated in urine cultures, which, in some cases, may require different antibiotic treatments than those suggested by urine cultures. For example, in one-third of the cases in this study, coincident identification of *E. faecalis* with Enterobacteriaceae was found, for which the choice of antibiotics differs between them.

This study has some limitations. Firstly, the population in our study corresponds to patients admitted to an internal medicine ward. Therefore, the applicability of the results could differ in other populations with UTIs, especially in patients admitted to the ICU or with nosocomial UTIs. Secondly, it was carried out at a single center, which may limit the generalizability of our findings.

The major strength of the current study is the rigorous selection of cases, including only those with complicated community-acquired UTIs, and excluding those patients in whom the diagnosis of UTI on admission was in doubt and the isolation of bacteria in urine could be due to asymptomatic bacteriuria. Our approach through clinical and epidemiological data, as well as microbiological data, offers another added strength, since our results are applicable in daily clinical practice. Lastly, the study’s prospective design and the homogeneity of its population are additional methodological strengths.

## 5. Conclusions

Our study found that febrile community-acquired UTIs and elevated procalcitonin were risk factors for bacteremia. Although the 30-day mortality in this study was almost double in patients with bacteremic UTIs compared to those without bacteremia, we could not find a statistically significant association between bacteremia and 30-day mortality, which was related to qSOFA and Barthel Index <40%.

## Figures and Tables

**Figure 1 microorganisms-11-01995-f001:**
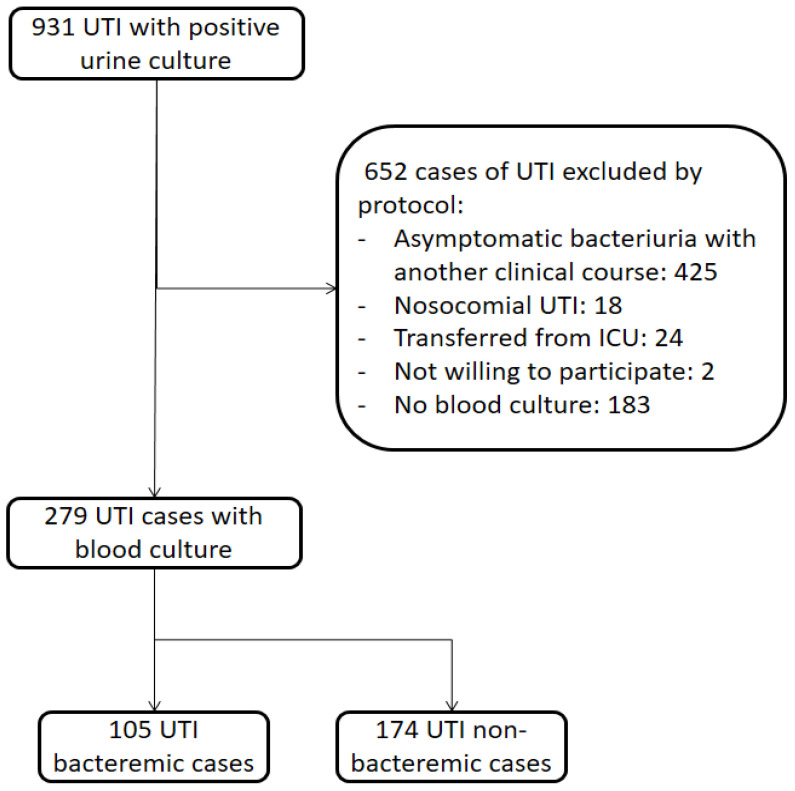
Flow chart of inclusion of 236 cases of complicated bacteremic and non bacteremic community-acquired urinary tract infection.

**Table 1 microorganisms-11-01995-t001:** Epidemiological and clinical characteristics and outcomes of bacteremic and non-bacteremic community-acquired urinary tract infections.

	Total N 279	Bacteremic UTI N 105	Non-Bacteremic UTI N 174	*p*
Female sex, *n* (%)	123 (44.1)	54 (51.4)	61 (40.7)	0.055
Age (years), median [IQR]	78 (70–85)	81 (72–87)	77 (67–83)	0.111
Charlson ≥ 5, *n* (%)	155 (55.6)	69 (65.7)	86 (49.4)	**0.008**
Barthel <40, *n* (%)	67 (24)	37 (35.2)	54 (31)	0.468
**Comorbidities**				
Dementia, *n* (%)	55 (22.9)	30 (28.6)	38 (24.4)	0.293
Diabetes mellitus, *n* (%)	89 (31.9)	37 (35.2)	52 (29.9)	0.353
COPD, *n* (%)	40 (14.3)	15 (14.3)	25 (14.5)	0.970
CKD, *n* (%)	62 (22.2)	37 (35.2)	52 (30.1)	0.369
Cancer, *n* (%)	54 (22.5)	22 (21)	40 (23)	0.692
Indwelling urinary catheter, *n* (%)	61 (21.9)	23 (21.9)	38 (21.8)	0.990
HCA-UTI, *n* (%)	154 (55.2)	62 (59)	92 (52.9)	0.315
Previous hospitalization, *n* (%)	89 (31.9)	37 (35.2)	52 (29.9)	0.353
Previous antimicrobial therapy, *n* (%)	133 (47.7)	50 (47.6)	83 (47.7)	0.989
Nursing home residence, *n* (%)	15 (5.4)	8 (7.6)	7 (4)	0.197
**Clinical characteristics**				
APN, *n* (%)	217 (77.8)	89 (84.8)	128 (73.6)	**0.029**
Altered mental status, *n* (%)	110 (39.4)	41 (39)	69 (39.7)	0.920
RR ≥ 22 bpm, *n* (%)	70 (25.1)	37 (35.2)	33 (19)	**0.002**
HR ≥ 90 bpm, *n* (%)	191 (68.5)	75 (71.4)	116 (66.7)	0.407
SBP <100 mmHg, *n* (%)	58 (20.8)	25 (23.8)	33 (19)	0.334
Temperature ≥ 38 °C, *n* (%)	132 (47.3)	62 (59)	70 (40.3)	**0.008**
qSOFA ≥ 2, *n* (%)	83 (29.7)	42 (40)	41 (23.6)	**0.004**
Sepsis (SOFA ≥ 2), *n* (%)	133 (47.7)	61 (58.1)	72 (41.4)	**0.007**
Septic shock, *n* (%)	40 (14.3)	22 (21)	18 (10.3)	**0.014**
qPitt ≥ 2, *n* (%)	115 (41.2)	40 (38.1)	75 (43.1)	0.410
Lactate ≥ 2 mg/dL	120 (43)	55 (52.4)	65 (37.4)	**0.014**
PCR ≥ 5 mg/L (%)	273 (99.6)	104 (99)	169 (97.1)	0.284
PCT ≥ 0.5 ng/mL, *n* (%)	87/99 (87.9)	37/40 (92.5)	50/59 (84.7)	**0.003**
Leukocytosis, median [IQR]	13,900 (10,400–18,500)	14,600 (10,400–18,600)	13,250 (10,500–18,400)	0.595
Polymicrobial UTI, *n* (%)	30 (10.8)	16 (15.2)	14 (8)	0.060
MDR, *n* (%)	89 (31.9)	37 (35.2)	52 (29.9)	0.353
ESBL, *n* (%)	33 (11.8)	15 (14.3)	18 (10.3)	0.323
IEAT, *n* (%)	67 (24)	23 (22.1)	44 (25.3)	0.550
**Etiology**				
*Escherichia coli*, *n* (%)	173 (56.5)	69 (57)	104 (56.2)	0.322
*Klebsiella pneumoniae*, *n* (%)	31 (10.1)	15 (12.4)	16 (8.6)	0.190
*Pseudomonas aeruginosa*, *n* (%)	24 (7.8)	9 (7.4)	15 (8.1)	0.989
*Enterococcus faecalis*, *n* (%)	24 (7.8)	8 (6.6)	16 (8.6)	0.649
*Proteus mirabilis, n* (%)	13 (4.2)	7 (5.8)	6 (3.2)	0.217

COPD, chronic obstructive pulmonary disease; CKD, chronic kidney disease; HCA-UTI, healthcare-associated-UTI; APN, acute pyelonephritis; RR, respiratory rate; SBP, systolic blood pressure; PCT, procalcitonin; MDR, multidrug-resistant; ESBL, extended spectrum beta-lactamase *Enterobacteriaceae*; IEAT, inadequate empirical antimicrobial therapy. *p* < 0.05 is considered statistically significant (in bold).

**Table 2 microorganisms-11-01995-t002:** Multivariate analysis of risk factors for bacteremia in complicated community-acquired urinary tract infections.

	Univariate Analysis		Multivariate Analysis	
	OR (95% CI)	*p*	OR (95% CI)	*p*
Charlson ≥ 5	1.5 (1.1–2.1)	0.008	1.6 (0.7–4.9)	0.276
Temperature ≥ 38 °C	1.3 (1.1–1.6)	0.008	1.3 (1.1–1.7)	**0.006**
qSOFA ≥ 2	1.6 (1.2–2.1)	0.004	0.8 (0.4–1.8)	0.462
Lactate ≥ 2 mg/dL	1.5 (1.1–1.9)	0.014	1.2 (0.6–3.3)	0.781
PCT ≥ 0.5 ng/mL	3.6 (−1.2–10.7)	0.003	8.5 (2.8–39.4)	**0.005**

PCT, procalcitonin. *p* < 0.05 is considered statistically significant (in bold).

**Table 3 microorganisms-11-01995-t003:** Univariate and multivariate analysis of risk factors for 30-day mortality of complicated community-acquired urinary tract infections.

	Univariate Analysis		Multivariate Analysis	
	OR (95% CI)	*p*	OR (95% CI)	*p*
Charlson ≥ 5	3.5 (1.6–7.8)	0.001	3.1 (0.8–10.8)	0.087
Barthel ≤40	6.7 (3.3–13.5)	<0.001	4.8 (1.3–18.2)	**0.020**
APN	0.7 (0.4–1.3)	0.283	-	-
HCA-UTI	2.3 (1.1–4.5)	0.014	1.8 (0.5–6.8)	0.370
Temperature ≥ 38 °C	1.9 (0.9–3.9)	0.051	-	-
PCT ≥ 0.5 ng/mL	4.1 (1.1–16.1)	0.019	2.2 (0.3–15.3)	0.423
Bacteremia	1.8 (0.9–3.3)	0.093	2.7 (0.7–10.3)	0.133
IEAT	1.3 (0.7–2.4)	0.452	-	-
qSOFA ≥ 2	12.6 (5.5–28.9)	<0.001	5.4 (1.2–24.9)	**0.030**
qPitt ≥ 2	3.9 (2.1–7.9)	<0.001	1.2 (0.3–5.1)	0.822

APN, acute pyelonephritis; HCA-UTI, healthcare-associated urinary tract infection; PCT, procalcitonin; IEAT, inadequate empirical antimicrobial therapy. *p* < 0.05 is considered statistically significant (in bold).

## Data Availability

The data presented in this study are available upon request from the corresponding author.

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
