# Peer review of "Risk Factors for Bacteremia and Its Clinical Impact on Complicated Community-Acquired Urinary Tract Infection"

_microorganisms, 2023, doi:10.3390/microorganisms11081995_

Round 1
Reviewer 1 Report
The urinary tract infections are second most important infections both in community and hospital healthcare. The presence of well-defined risk factors for worse outcome of community acquired urinary tract infections associated with bacteraemia should be helpful. Therefore, the presented topic is actual. But in presented text I have some problems; some parts are by my opinion out of paper title.
E.g.
Both Charleson and Barthel index have only minimal association with urinary tract (the evaluation of voluntary micturition in Barthel or chronic renal disease in Charleson), therefore their use in evaluation of urinary tract infection with mortality is incorrect, because the co-morbidities itself are the reason for worse outcome and not the UTI.
The same – qSOFA or qPitt have been routinely used for evaluation of sepsis severity and again both are markers of organism overall alteration independent to urinary tract infection. There is no logic reason for their use in paper evaluate the association of UTI with patient’s outcome.
Minor problems:
1. Article name – rewrite “ITS” to “its”
2. Table 1 – for better clarity I suggest to highlight the statistically significant results, the bold font is not enough
Author Response
Reviewer 1
We thank the reviewer for their comments, and below we address each of them, indicating the modifications we have made, which we believe have improved the manuscript.
The urinary tract infections are second most important infections both in community and hospital healthcare. The presence of well-defined risk factors for worse outcome of community acquired urinary tract infections associated with bacteraemia should be helpful. Therefore, the presented topic is actual. But in presented text I have some problems; some parts are by my opinion out of paper title.
E.g.
Both Charleson and Barthel index have only minimal association with urinary tract (the evaluation of voluntary micturition in Barthel or chronic renal disease in Charleson), therefore their use in evaluation of urinary tract infection with mortality is incorrect, because the co-morbidities itself are the reason for worse outcome and not the UTI.
Answer: We appreciate the reviewer's comments; however, we believe that their suggestion not to use Charlson's index and the Barthel index is not applicable to this study. These indices are applied in clinical research on various severe infections, including urinary focus. We hope that the arguments we present below will be pleasing to the reviewer.
The Charlson Comorbidity Index and the Barthel index are two widely used scores for comorbidity and dependency, respectively. Both scores have been used in studies of cardiovascular diseases, aging and infectious diseases from different sources. In our study, we had an old population (median age 78 [70-85] years), and we used these scores to evaluate the impact of comorbidity and dependency in both mortality and bacteriemia. None of them was related to bacteremia, as could be, correlating comorbidity and dependence with immunosenescence. Nonetheless, as expected, dependency was associated with 30-day mortality.
The same – qSOFA or qPitt have been routinely used for evaluation of sepsis severity and again both are markers of organism overall alteration independent to urinary tract infection. There is no logic reason for their use in paper evaluate the association of UTI with patient’s outcome.
Answer: Sepsis is a systemic process of life-threatening organic dysfunction; however, the characteristics of the patient, the microorganism, and the source of infection influence the prognosis. Quick SOFA has been widely used as a prognostic score in sepsis since it was developed in 2016 with three variables related to in-hospital mortality and a combined variable of mortality and ICU stay of more than 72 hours in patients with suspected infection (Seymour, JAMA 2016). In that study the sources of the infections are not specified; however, in other subsequent studies it has been in the hospital ward in patients with urinary tract infection (Madrazo et al. International Journal of Clinical Practice 2021), suggesting a greater accuracy than with respect to sepsis from other infections (Estella et al., European Journal of Internal Medicine 2018). Quick Pitt is not a routinely used score, as it was developed and validated by Battle et al. in 2019 to simplify the more used Pitt score. In the study in which the score is developed Battle et al. Infection 2019), it evaluates the risk of 14-day mortality in bloodstream infections due to gram-negative bacilli from different sources, being urinary tract infection the most frequent focus (53% of the cases). In our opinion there is no reason not to use these scores to evaluate prognosis in patients with complicated urinary tract infection, that may develop sepsis (almost half of the patients in our study -47.7%- had sepsis criteria at admission).
Therefore, we believe that the use of these indices is justified by previous studies in infectious pathology, which include patients with UTI and, furthermore, add value to the assessment of the impact of bacteremia on the prognosis in UTI in this study. We hope that the reviewer, considering the characteristics of the population included in our work, will positively evaluate the use of these prognostic indices.
Minor problems:
- Article name – rewrite “ITS” to “its”
Answer: Thank you for the correction. It has been done. The Title now reads: “Risk Factors for Bacteremia and its Clinical Impact on Complicated Community-acquired Urinary Tract Infection” (see lines 2-3).
- Table 1 – for better clarity I suggest to highlight the statistically significant results, the bold font is not enough
Answer: We appreciate your comment. While we have complied with the reviewer’s instructions in creating the tables, we have enlighted the significant results of tables 1, 2 and 3 to enhance the tables’ readability, following your indications.

Reviewer 2 Report
This is a well-conducted prospective study that included complicated community-acquired UTI and evaluated those patients who developed bacteriemia vs. those who did not. Moreover, they performed a multivariate analysis, including risk factors for bacteriemia and 30-day mortality. They found that fever and elevated procalcitonin were related to bacteremia, and qSOFA and Barthel index <40% were associated with mortality.
Considerations
In the abstract, the sentence:
"Bacteremia has been associated with severity in some infections; however, its clinical impact on urinary tract infection." The reading seems that bacteremia impacts UTIs and should be the opposite.
Introduction
The introduction has a paragraph related to asymptomatic bacteriuria, which differs from this study's subject. I suggest putting this criterion in the methodology as an exclusion criterion and a brief definition.
Methods
Was there any patient with a urinary catheter other than a Foley catheter, such as a JJ catheter or nephrostomy?
A non-specific inflammatory marker that may also be useful is C-reactive protein. Is there any particular reason for not including it?
It is not helpful to break down the Charlson scale or the diagnostic criteria for DM2, COPD, or some other comorbidities, considering that they are already patients with these diagnoses previously made, and indeed many of them with treatment. Moreover, this reduces the size of the manuscript.
Include exclusion criteria (those specified in the figure: asymptomatic bacteriuria, nosocomial, etc.)
Results
Moreover, were there isolates (in blood or urine) with S. aureus or Candida? Considering these pathogens that could be important in this type of infection.
Author Response
Revisor 2
This is a well-conducted prospective study that included complicated community-acquired UTI and evaluated those patients who developed bacteriemia vs. those who did not. Moreover, they performed a multivariate analysis, including risk factors for bacteriemia and 30-day mortality. They found that fever and elevated procalcitonin were related to bacteremia, and qSOFA and Barthel index <40% were associated with mortality.
Considerations
In the abstract, the sentence:
"Bacteremia has been associated with severity in some infections; however, its clinical impact on urinary tract infection." The reading seems that bacteremia impacts UTIs and should be the opposite.
Answer: Thank you for your comment. We have rewritten the sentence, and it now reads: "Bacteremia has been associated with severity in some infections, however, its impact on the prognosis of urinary tract infection is still disputed” (lines 10 and 11).
Introduction
The introduction has a paragraph related to asymptomatic bacteriuria, which differs from this study's subject. I suggest putting this criterion in the methodology as an exclusion criterion and a brief definition.
Answer: We appreciate your comment. Following your recommendations, we have removed the paragraph related to asymptomatic bacteriuria from the introduction (lines 51-56) and we have added it to the Material and Methods section (lines 74-77). We have also improved the description of the exclusion criteria (lines 77-78).
Methods
Was there any patient with a urinary catheter other than a Foley catheter, such as a JJ catheter or nephrostomy?
Answer: Thank you for your question. Most of the urinary catheter were Foley catheter, but there were others. A paragraph has been added to Results section to better explain this point (lines 284-288)
A non-specific inflammatory marker that may also be useful is C-reactive protein. Is there any particular reason for not including it?
Answer: Thank you for your comment. C-reactive protein is a non-specific inflammatory marker usually elevated in infections. However, it was not useful to discern between bacteremic and non-bacteremic infections in this study. Following your suggestions we included it in the preliminary analysis (see Table 1).
It is not helpful to break down the Charlson scale or the diagnostic criteria for DM2, COPD, or some other comorbidities, considering that they are already patients with these diagnoses previously made, and indeed many of them with treatment. Moreover, this reduces the size of the manuscript.
Answer: We appreciate your comments. We agree with your idea that shortening the manuscript can enhance its readability. However, following the editorial guidelines, we had attempted to provide maximum detail in the methodology to make it accessible to readers. Nevertheless, based on your instructions, we have removed the paragraph detailing the Charlson Comorbidity Index. (lines 153-164).
Include exclusion criteria (those specified in the figure: asymptomatic bacteriuria, nosocomial, etc.)
Answer: Thank you for your suggestion. As previously indicated, we have added the paragraph about asymptomatic bacteriuria to the Material and Methods section (lines 74-77), and we have provided a more detailed description of the exclusion criteria (lines 77-78).
Results
Moreover, were there isolates (in blood or urine) with S. aureus or Candida? Considering these pathogens that could be important in this type of infection.
Answer: We appreciate your comments. The isolation of S. aureus and Candida in our study were very scarce, however, as these pathogens could be significant, we have included one paragraph in the Results section (lines 317-322) and we have written the names of the microorganisms in italics.
Sincerely,
Juan Alberola
